# Microbial Degradation, Spectral analysis and Toxicological Assessment of Malachite Green Dye by *Streptomyces exfoliatus*

**DOI:** 10.3390/molecules27196456

**Published:** 2022-09-30

**Authors:** Samah H. Abu-Hussien, Bahaa A. Hemdan, Othman M. Alzahrani, Amal S. Alswat, Fuad A. Alatawi, Muneefah Abdullah Alenezi, Doaa Bahaa Eldin Darwish, Hanouf S. Bafhaid, Samy F. Mahmoud, Mohamed F. M. Ibrahim, Salwa M. El-Sayed

**Affiliations:** 1Department of Microbiology, Faculty of Agriculture, Ain Shams University, Cairo 11566, Egypt; 2Environmental and Climate Change Research Institute, National Research Centre, Giza 1266, Egypt; 3Department of Biology, College of Science, Taif University, P.O. Box 11099, Taif 21944, Saudi Arabia; 4Department of Biotechnology, College of Science, Taif University, P.O. Box 11099, Taif 21944, Saudi Arabia; 5Biology department, Faculty of science, Tabuk University, Tabuk 71491, Saudi Arabia; 6Botany Department, Faculty of science, Mansoura University, Mansoura 35511, Egypt; 7Pharmacology and Toxicology Department, College of Pharmacy, Umm Al Qura University, Makkah 24381, Saudi Arabia; 8Department of Agricultural Botany, Faculty of Agriculture, Ain Shams University, Cairo 11566, Egypt; 9Department of Biochemistry, Faculty of Agriculture, Ain Shams University, Cairo 11566, Egypt

**Keywords:** *Streptomyces exfoliatus*, biodegradation, decolorization, malachite green dye, phytotoxicity, response surface methodology, cytotoxicity, gas chromatography-mass spectrometry GC-MS

## Abstract

Malachite green (MG) dye is a common environmental pollutant that threatens human health and the integrity of the Earth’s ecosystem. The aim of this study was to investigate the potential biodegradation of MG dye by actinomycetes species isolated from planted soil near an industrial water effluent in Cairo, Egypt. The *Streptomyces* isolate St 45 was selected according to its high efficiency for laccase production. It was identified as *S. exfoliatus* based on phenotype and 16S rRNA molecular analysis and was deposited in the NCBI GenBank with the gene accession number OL720220. Its growth kinetics were studied during an incubation time of 144 h, during which the growth rate was 0.4232 (µ/h), the duplication time (td) was 1.64 d, and multiplication rate (MR) was 0.61 h, with an MG decolorization value of 96% after 120 h of incubation at 25 °C. Eleven physical and nutritional factors (mannitol, frying oil waste, MgSO_4_, NH_4_NO_3_, NH_4_Cl, dye concentration, pH, agitation, temperature, inoculum size, and incubation time) were screened for significance in the biodegradation of MG by *S. exfoliatus* using PBD. Out of the eleven factors screened in PBD, five (dye concentration, frying oil waste, MgSO_4_, inoculum size, and pH) were shown to be significant in the decolorization process. Central composite design (CCD) was applied to optimize the biodegradation of MG. Maximum decolorization was attained using the following optimal conditions: food oil waste, 7.5 mL/L; MgSO_4_, 0.35 g/L; dye concentration, 0.04 g/L; pH, 4.0; and inoculum size, 12.5%. The products from the degradation of MG by *S. exfoliatus* were characterized using high-performance liquid chromatography (HPLC) and gas chromatography-mass spectrometry (GC-MS). The results revealed the presence of several compounds, including leuco-malachite green, di(tert-butyl)(2-phenylethoxy) silane, 1,3-benzenedicarboxylic acid, bis(2-ethylhexyl) ester, 1,4-benzenedicarboxylic acid, bis(2-ethylhexyl) ester, 1,2-benzenedicarboxylic acid, di-n-octyl phthalate, and 1,2-benzenedicarboxylic acid, dioctyl ester. Moreover, the phytotoxicity, microbial toxicity, and cytotoxicity tests confirmed that the byproducts of MG degradation were not toxic to plants, microbes, or human cells. The results of this work implicate *S. exfoliatus* as a novel strain for MG biodegradation in different environments.

## 1. Introduction

Throughout many regions of the world, surface water quality is a significant environmental problem. This is a paramount concern that has an effect on public health and natural systems. Surface and groundwater quality is negatively impacted by anthropogenic sources—such as urbanization, agricultural and industrial operations, chemical waste disasters, and dam construction—and natural procedures, including emission and weather conditions [1]. Indeed, as the process of urbanization advances, the global impacts grow increasingly apparent day by day [2]. 

Waste dumping is becoming more harmful to the environment, and new technologies for the bioremediation of hazardous chemicals must be established to lessen these negative consequences. During the textile dying process, up to 50% of dyes are not absorbed by the fibers and remain as pollutants, posing a considerable environmental risk. The discharge of dyes produces an intense color even in small concentrations due to their turbidity and high pollution intensity. Dyes can significantly impact the aquatic environment through their toxic degradation products [3]. 

Due to the wide variation in the chemical composition of these molecules, it is extremely difficult to determine specific traditional, chemical, or biological techniques that can be used to remediate the damage caused by dumping these hazardous wastes, despite the adverse effects of dye effluents discharged by the textile and dyeing industry on water bodies and increasing public concern about their toxicity and carcinogenicity. As a consequence, innovative production technologies must be investigated. Azo dyes account for half of the dyes used in the textile industry, and releasing these products into the environment can cause related pollution problems [4]. Textile effluent can be treated using a variety of techniques that decolorize dyes via physical, chemical, or biological treatments [5].

MG is a triphenylmethane dye widely used in the silk dyeing, aquaculture, and textile industries and has been reported to be toxic to all organisms. It is considered a common environmental pollutant that poses a significant threat to non-target organisms, including humans. Moreover, due to its reported toxic effects, the dye has generated a great deal of concern regarding its use [6]. The toxicity of this dye increases with exposure time, temperature, and concentration. It has been reported to cause carcinogenesis, mutagenesis, chromosomal fractures, teratogenicity, and respiratory toxicity [7].

As reported by the FAO [8], MG is a common triphenylmethane dye that is widely used in the textile and dyeing industries. This compound has also long been applied as a fishery medicine in China’s aquaculture industry because it controls certain diseases, such as saprolegniasis. However, MG residues in aquaculture products and the environment may threaten human health, and the potential teratogenic, carcinogenic, and mutagenic effects of MG have been reported since the 1970s [8]. Nevertheless, malachite green is still used illegally because of its low price and good sterilization effect. In addition, thousands of tons of wastewater from triphenylmethane dye production are discharged into rivers and lakes, and its residues persist in soils. Triphenylmethane dye residues then enter the food chain, thus considerably threatening human health. Therefore, developing an efficient method for MG degradation is crucial, as discussed by Bottoni and Caroli [9].

In microbial biodegradation assays, heterotrophic bacteria, such as *E. coli* and *Pseudomonas luteola*, and fungal strains, including *Aspergillus niger*, have been used to remove dyes from many environments [10]. Nonetheless, the impact of dye degradation by actinomycetes has not been thoroughly investigated. Vignish et al. [11] reported that extracellular enzymes produced by actinomycetes can help to dissolve intractable compounds, giving them an advantage over single-celled organisms. Actinomycetes are considered a highly abundant and metabolically diverse group with the ability to produce various secondary metabolites that are used in many therapeutic applications [12]. Moreover, actinomycetes are well known for their beneficial impact on the biodegradation of hazardous compounds [13]. In addition, the Actinobacteria phylum was also recognized for being capable of surviving in extreme environments and tolerating high concentrations of toxins [11]. Thus, actinomycetal strains could be a promising solution for the bioremoval or biodegradation of harmful dyes from water and soil effluents. The main objectives of the present study were: (1) to degrade MG dye using *S. exfoliatus* by applying the response surface methodology (RSM) statistical approach; (2) to determine the kinetic parameters of MG decolorization by *S. exfoliatus*; and (3) to identify the intermediate compounds produced during MG biodegradation via spectral analysis as well as study the microbial toxicity, phytotoxicity, and cytotoxicity of MG byproducts.

## 2. Results

### 2.1. Isolation of MG-Biodegrading Actinomycetes Isolates

Twenty soil samples were collected and prepared for actinomycetes isolation on casein starch nitrite agar supplemented with nalidixic acid (20 μg/mL) and nystatin (100 μg/mL) to inhibit the bacteria and fungi, respectively, and incubated at 25 °C for 5–7 days. All isolates were then tested for laccase production on Vogel’s mineral media (VMM) agar plates. One hundred and twenty-eight isolates were obtained from the soil samples on casein starch nitrate agar. Among them, 85 different actinomycetes showing laccase enzyme production were separated on the basis of pigmentation (Figure 1). Actinomycetes strains producing white and black pigments were most predominant at 31% and 23%, respectively. Eighty-five isolates were positive for laccase production. *Streptomyces* St 45 was the most efficient isolate; hence, it was selected for further studies. 

### 2.2. Molecular Identification of Streptomyces St45 Isolates Using 16S rRNA Genetic Sequencing

The classification of dye-degrading *Streptomyces* spp. relied on 16S rRNA gene sequence analysis, which provided the most knowledge regarding *Streptomyces* and could be exploited to establish the many unique species of *Streptomyces* strains. The neighbor-joining algorithm was employed to assemble the phylogenetic tree depicted in (Figure 2) from the distance measures. In the phylogenetic tree, the overwhelming majority of sequences established groupings. However, we observed the existence of various *Streptomyces* 16S rRNA sequence types, alluding to several unique *Streptomyces* species. An unknown isolate that showed MG biodegradation activity was picked and sequenced using 16S rRNA sequencing analysis. The results illustrated in Figure 2 showed that the 16S rRNA gene sequence was consistent with that of *S. exfoliatus* (NCBI, Bethesda, MD, USA); thus, it was identified as *S. exfoliatus* (accession No. OL720220). Furthermore, the affinity between the isolates and their nearest phylogenetic neighbors is shown in Figure 2, which includes a comparison of the sequence information for the several Streptomyces isolates. The *Streptomyces* 16S rRNA gene tree included many groups in other sequence categories; however, the phylogenetic branches were generated from a multitude of sequences. The findings revealed an almost 95% sequence similarity between the 13 *Streptomyces* spp. and *S. exfoliatus*.

### 2.3. Optimization of MG Dye Decolorization Using Statistical Experimental Strategies

#### 2.3.1. Effect of Incubation Time on the Decolorization of MG Dye

The effect of incubation time on MG decolorization was studied using *S. exfoliatus* and an incubation period of 120 h at 30 °C. As shown in (Figure 3 and Figure 4), MG degradation started after 72 h of incubation and reached 99% at the end of 120 h. The high R^2^ coefficient indicated that incubation time significantly affected MG removal (98%) with a specific growth rate of 0.3055 µ/h, duplication time (td) of 2.27 d, and multiplication rate (MR) of 0.44 h during the logarithmic phase.

#### 2.3.2. Screening of Physical and Nutritional Factors Affecting Dye Removal by *S. exfoliatus* Using Plackett–Burman Design (PBD)

PBD was used to screen the most significant physical and nutritional factors affecting the decolorization process over 18 experimental runs at high and low levels for each factor. As shown in Table 1, run 18 (mannitol, 5.00 g/L; frying oil waste, 5.00 mL/L; magnesium sulphate, 0.25 g/L, ammonium nitrate, 1.00 g/L; ammonium chloride, 1.00 g/L; malachite green dye, 0.05 g/L; pH, 5; inoculum size, 10%; incubation temperature, 30 °C; and incubation period, 120 h) had the highest degradation percentage (81.1%). The *F*-value model was 18.17 with low probability (*p* = 0.004), where a high *F*-value and *p* > *F* > 0.05 indicate significant model terms. As seen in (Figure 5), mannitol, frying oil waste, magnesium sulfate, agitation speed, temperature, inoculum size, and incubation period were significant at their higher values. In contrast, ammonium nitrate, ammonium chloride, dye concentrations, and pH were significant at their lower values.

The standard deviation was 5.41 and the determination coefficient (R^2^) was 0.96, indicating that the model described 96% of the total variance; this suggests a good characterization of the process model and a strong correlation. Thus, the linear equation that describes the % dye decolorization (Y) response can be expressed as follows:Y (decolorization) = + 29.03 + 2.45 A + 4.25 B + 6.46 C - 2.88 D − 2.54 E − 5.30 F + 3.19 H + 8.73 J + 3.54 K − 15.33 L
where A is mannitol, B is frying oil waste, C is magnesium sulfate, D is ammonium nitrate, E is ammonium chloride, F is the pH level, G is the agitation speed, H is the temperature, J is the inoculum size, K is the incubation period, and L is the dye concentration.

#### 2.3.3. Optimization of the Most Significant Factors for Dye Decolorization Using Composite Central Design (CCD)

In this study, 32 experiments with different combinations of frying oil waste, magnesium sulfate, dye concentrations, pH, and inoculum size were performed at three different levels (low, moderate, and high). As shown in Table 1, the maximum decolorization percentage (98.10%) was achieved in trial run 3 with the following parameters: 7.5 mL/L of frying oil waste, 0.35 g/L of magnesium sulfate, pH 4.0, an inoculum size of 12.5%, and a dye concentration of 0.04 g/L. The variance analysis showed that the model’s *F*-value was 5.00 with a low *p*-value of 0.0004, indicating the significance of the model. Likewise, significant interactions between the dye concentration and pH and magnesium sulfate and pH were recorded for dye decolorization (*p* = 0.005 and 0.001, respectively), while the other parameter interactions were not meaningful. The determination coefficient (R^2^) was 0.90, indicating that the model could explain 90% of the variability in the response and exhibited a high correlation. The standard deviation was determined as 1.74.

The main effects, interactions, and quadratic effects can be described in the final equation as follows:Y (decolorization%) = −1.90902 + 2274.8 A + 2.8 B + 47.3 C + 17.4 D − 2.5 A − 23.6 A∗E − 411.3 A∗C − 105.5 A∗D + 19.2 A∗B − 0.6 E∗C − 0.2 E∗D + 0.05 E∗B − 9.97 C∗D + 3.4 C∗B + 0.3 D∗A − 18282.9 A^2^ − 0.2 E^2^ − 18.8 C^2^
− 1.0 D^2^ – 0.2 B^2^. 
where Y is the predicted response (decolorization%), A is the dye concentration (mg/L), B is frying oil waste (%), C is magnesium sulfate (g/L), D is the pH level, and E is the inoculum size (mL/L). The use of multi-factor design to optimize the biochromic process in dyes is essential to achieve the highest possible color change rate. It can be seen from this design that several factors have a great influence on the biological discoloration process.

In Figure 6, the regression equation is represented in 3D response surface plots, and 2D contour plots were also plotted to determine the optimum concentration of each factor for maximum dye decolorization. The contour plots were elliptical, indicating that the pH–dye concentration and pH–magnesium sulfate interactions were significant, given that only the meaningful interactions were plotted. Regarding the effect of pH on the discoloration process, it was found that the pH range of the fungi tested for the growth and discoloration of MG dyes was extensive. The optimal pH values of *S. exfoliatus* were 6 and 7.

### 2.4. Phytotoxicity

The effect of malachite green dye and its degradation metabolites on the germination percentage (%) of broccoli (*Brassica oleracea var. italic*) and lettuce (*Lactuca sativa* var. *capitata*) seeds was studied in comparison to a distilled water (control) treatment. According to the recorded data, the control treatment had the highest germination rate, at 96% and 98% for the broccoli and lettuce seeds, respectively. The lowest germination percentage was found for malachite green, indicating that it had inhibitory effects on the seeds, with germination rates of 60% and 65% for the broccoli and lettuce seeds, respectively. Compared to the control treatment, the degraded MG metabolites caused a non-significant increase in broccoli and lettuce germination, reaching 90% and 94%, respectively, as shown in Figure 7. Similarly, the effect of MG degradation products was studied in *V. faba* 6 weeks after planting. As shown in Figure 8, no significant toxic effects of MG degradation products were observed on shoot length, root length, and the number of leaves when compared to the control treatments, reaching 10.33, 7.11, and 7, respectively, for plants irrigated with MG before degradation; 17.33, 10.33, and 14, respectively, for plants treated with MG products after degradation; and 17.5, 10.22, and 14, respectively, for the control treatments.

### 2.5. Microbial Toxicity

The well diffusion method was used to test for MG toxicity in some microbial strains, as described by the CLSi, 2015. As shown in Figure 9, the MG products showed no toxicity in all bacterial and fungal strains (*S. aureus* ATCC 29737, *E. coli* ATCC 8379, *C. albicans* ATCC 60193, and *A. niger* ATCC 16404) studied, as no inhibition zone was recorded.

### 2.6. Cytotoxicity

The effects of the biodegraded MG products were studied in normal HSF cells using an MTT assay at ten concentrations from 0 to 200 µg/mL. As shown in Figure 10, there was no toxicity observed at both the low and high concentrations in normal HSF cells. At the lower concentration of 1 µg/mL, the HSF cells showed 97% cell viability, compared to 96% at an MG byproduct concentration of 200 µg/mL. An HSF cell viability of 100% was recorded for the control treatment. GraphPad Prism (5) was used to calculate the half-maximal inhibitory concentration (IC_50_), which was >200 µg/mL. Microscopic images illustrated no increased cytotoxicity for MG byproducts when compared to the control treatment.

### 2.7. GC-MS Analysis for MG Biodegradation Products

The loss of MG color throughout incubation indicated that the dye was converted to its leuco- form (Figure 3A and 8). From the results of HPLC, we obtained a clearly distinguishable specific peak corresponding to an intermediate leuco-malachite green (LMG) candidate product. This peak was found at a retention time of 10.0 min (Figure 11). However, we still lacked enough deterministic data to confirm the categories of these compounds. As a result, GC-MS was employed, and several probable intermediate products were identified (Figure 12).

To confirm this hypothesis, the metabolites from resting *S. exfoliatus* cells were analyzed using GC-MS to detect the intermediate products. From the results of GC-MS, we obtained several specific peaks for candidate intermediate products that were clearly distinguishable, including di(tert-butyl)(2-phenylethoxy)silane; 1,3-benzenedicarboxylic acid, bis(2-ethylhexyl) ester; 1,4-benzenedicarboxylic acid, bis(2-ethylhexyl) ester; 1,2-benzenedicarboxylic acid; di-n-octyl phthalate; and 1,2-benzenedicarboxylic acid, dioctyl ester. MG was not identified by GC-MS, indicating that the strain exhibited a high biodegradation efficiency.

## 3. Discussion

Despite the fact that textile industries have contributed to global economic development, their effluents are considered a significant cause of water pollution worldwide. As reported by Peryasamy et al. [14], the world produces 1,000,000 tons of coloring dyes annually. Out of these million tons, 280,000 tons contaminate textile effluents each year. It is well known that these dyes are non-degradable, and their presence can lead to a reduction in sunlight in aquacultures, inhibiting the photosynthesis process and decreasing the abundance of all aqua flora, as reported by Berradi et al. [15]. MG dye is used in the aquaculture industry as a parasiticide, as it controls fungal infections, protozoan attacks, and disease caused by helminths. Thus, this dye has unique and necessary uses. However, it plays an essential role in causing cancers, cell mutations, chromosomal fractures, and toxic respiratory effects. It is well known that MG dye toxicity increases with exposure time, temperature, and dye concentration. Toxicity occurs in some mammals and can be observed as organ damage and mutagenic, carcinogenic, and developmental abnormalities. However, despite the large number of data on its toxic effects, MG dye is still used as a parasiticide in aquaculture and other industries. Many studies have confirmed the need for the removal of residual MG in treatment ponds. In our study, *S. exfoliatus* was shown to have the ability to produce the laccase enzyme. Laccases (EC 1.10.3.2; benzenediol: oxygen oxidoreductase) are a family of multicopper oxidases that bear a close resemblance to ascorbate oxidase and mammalian plasma protein ceruloplasmin. Dye-degrading microorganisms have the ability to produce enzymes to hydrolyze dyes, such as azo-reductase, manganese-lignin peroxidase, and ascorbate oxidase, indicating their prominent roles in dye degradation. The role of laccase and peroxidase produced by Actinobacteria in the biodegradation of MG dye was reported by Shanmugam et al. [16], who showed that they produce an extracellular oxidoreductive, nonspecific, and non-stereoselective enzyme system including lignin peroxidase, tyrosinase, manganese peroxidase, and laccase to destroy MG dye.

Regarding the effect of different carbon sources on the bleaching process, it was discovered that frying oil waste—except for mannitol, which is a poor carbon source—could grow and bleach malachite green well. When frying oil waste was employed, the growth rate observed was rapid. The data revealed that the proportion of mannitol discoloration did not differ significantly. Ali et al. [17] found that using sucrose at an initial concentration of 10 g/L resulted in maximum decolorization after 96 h of incubation in the presence of 300 mg/L of MG dye. In one study, *Kocuria rosea* MTCC 1532 was used to decolorize malachite green using molasses and sucrose as carbon sources (91%) [18]. The necessity for a carbon source varies depending on the organism and the dye to be treated. Under all nitrogen sources, *S. exfoliatus* demonstrates no substantial growth and staining of malachite green dye. Parshetti et al. [19] found that the inclusion of peptone in the growth medium inhibited the staining of reactive blue 25 by *Aspergillus ochre*. Regarding the color change procedure, the culture of the investigated fungal isolates is usually superior to shaking conditions in terms of dye removal and fungal growth. Knapp et al. [20] and Revankar et al. [21] discovered that the dye discoloration in the corrected culture was due mainly to dye adsorption in the fungal mat, while no dye adsorption was observed in the fungal mycelium in the shaking culture, indicating that the tested fungus was a filamentous fungus. Lacina et al. [22] reported that *Tenacibaculum* spp. HMG1 could decolorize 98.8% of 20 mg/L malachite green in 12 h in LB medium. Taoufik et al. [2] reported that *Pseudomonas* spp. YB2 could almost completely decolorize 1000 mg/L MG within 12 h in the LB medium. However, achieving similar degradation efficiencies in the natural environment presents difficulties. Another important property of *P. veronii* JW3-6 is that it can degrade malachite green and other triphenylmethane dyes over a wide range of temperatures (20–40 °C) and pH levels [23,24,25,26]. These results demonstrate the potential of this microorganism for use in biodegradation and present new opportunities for its future applications. MG dye is removed via biodegradation and/or biosorption. Our strain was observed to grow and degrade MG (100 mg/L) at pH ranges between 6 and 8.0. The growth kinetics were studied for an incubation time of 144 h, reaching a growth rate of 0.4232 (µ/h), duplication time (td) of 1.64 d, and multiplication rate (MR) of 0.61 h with 96% MG decolorization after 120 h of incubation at 25 °C. Eleven physical and nutritional factors (mannitol, frying oil waste, MgSO_4_, NH_4_NO_3_, NH_4_Cl, dye concentration, pH, agitation, temperature, inoculum size, and incubation time) were screened for the *S. exfoliatus* MG biodegradation process using PBD. Out of the eleven factors screened, five (dye concentration, frying oil waste, MgSO_4_, inoculum size, and pH) were shown to be significant in the decolorization process. Central composite design (CCD) was applied to optimize the biodegradation of MG. Maximum decolorization was attained using the following optimal conditions: food oil waste, 7.5 mL/L; MgSO_4_, 0.35 g/L; dye concentration, 0.04 g/L; pH, 4.0; and inoculum size, 12.5%. Similarly, Angamuthu et al. [17] found that strain S20 was found to grow and decolorize MG (300 mg/L) at pH values between 5.0 and 9.0. However, the maximum degradation of 98.33  ±  0.17% of MG (300 mg/L) was observed at pH 7, while 94  ±  0.5% degradation was observed at pH 8 after a 96 h incubation period. In addition, Pradnya et al. [27] reported that *Enterococcus* sp. decolorized 94% of MG when used at a concentration of 20 mg/L at pH 7, while 90% degradation was observed at pH 8. Only 71  ±  0.5% degradation was shown at pH 5 due to the acidic conditions (Figure 4C). Alshehrei [28] reported only 16.2  ±  1.6% MG decolorization by *Bacillus cereus* and 19.3  ±  1.2% decolorization by *Pseudomonas aeruginosa* in an acidic environment.

Seed germination is the most efficient assay to assess the phytotoxicity of any end products for various bioremediation processes, as reported by Kapanen et al. [29]. In our study, broccoli and lettuce seeds were treated with biodegraded MG dye, which was shown to have no toxic effect on the seed germination results. Similarly, Angamathu et al. [17] found that mung bean (Vigna radiata) seeds treated with biodegraded MG products from *S. chrestomyceticus* S20 showed a significantly higher percentage of seed germination, reaching 92  ±  0.8%. 

Here, the biodegraded MG products were studied for their toxicity potential against some bacterial and fungal strains. The results showed no toxic effect of the biodegraded MG products, as no zone of inhibition was recorded around wells filled with the products. These results are in line with those of Sneha et al. [30], who used a well diffusion assay and found an absence of inhibition zones due to the nontoxicity of MG biodegradation products in *B. cereus*, *S. aureus*, *M. luteus*, *Streptococcus* sp., and *E. coli*. These results suggest that effluent containing MG would not be lethal to microflora after treatment with the *S. exfoliatus* strain.

This study was concerned with the cytotoxicity of the end products of MG biodegradation, and the cytotoxicity assay showed that the products of MG decolorized by *S. exfoliatus* had less of a toxic effect than untreated MG dye (control) when tested on the MCF7 cell line, as illustrated in Figure 9. The *S. exfoliatus*-treated samples showed 97% and 96% cell viability in the MCF-7 cell line (compared to 100% for the untreated dye sample) at concentrations of 1 and 200 µg/mL, respectively. Our results indicated that treated MG products had minimal toxicity in human cells when compared to the untreated MG dye solution. According to the results of Bhavsar et al. [31], the MG degradation products of laccase-decolorized dye samples were shown to have low cytotoxicity in MCF-7 cells and the normal human lung epithelial cell line (L132) when compared to the untreated sample. Indeed, the results of this work indicate that *S. exfoliatus* has a high potential for the degradation of MG in various environments; in addition, the products of this degradation are nontoxic to humans. 

According to the overall results and earlier studies, the decolorization performance is dependent on both the composition of the dye and the induced condition of all the degrading enzymes during the metabolic processes, as well as the microorganism used in the dye decomposition [32]. In the current investigation, the involvement of the strain *S. exfoliatus* in the decolorization of MG suggested a new metabolic pathway. Thus, we proposed a probable MG degradation pathway for the *S. exfoliatus* strain based on the identified intermediates, as illustrated in Figure 13. 

In this pathway, MG is reduced to form leuco-malachite green in the first step, which is then oxidized, and a heterocyclic ring is formed. *S. exfoliatus* then produces a new degradation product—di(tert-butyl)(2-phenylethoxy)silane (retention time: 4.05)—by the further reduction in LMG (Figure 10). Then, di(tert-butyl)(2-phenylethoxy)silane is converted to 1,3-benzenedicarboxylic acid, bis(2-ethylhexyl) ester (retention time: 36.52) or 1,4-benzenedicarboxylic acid, bis(2-ethylhexyl) ester (retention time: 36.52) through the cleavage of the heterocyclic ring and deamination, which are then converted to 1,2-benzenedicarboxylic acid by demethylation and treated to form 1,2-benzenedicarboxylic acid, di-n-octyl phthalate and 1,2-benzenedicarboxylic acid, dioctyl ester. These were the end products discovered in this study. According to previous studies, the first stage in MG biodegradation is either a direct stepwise demethylation process or a reduction reaction followed by a stepwise demethylation process [33,34,35]. The degradation mechanism of MG by bacteria usually starts with a reduction reaction followed by a stepwise demethylation process, whereas the degradation pathway of MG by fungi usually starts with a direct stepwise demethylation process [33,34,35,36]. Malachite green and its reduced form LMG are both extremely toxic [37]. There is concern about the fate of MG and LMG in aquatic and terrestrial ecosystems, where they appear as pollutants. The most frequently used techniques for assessing phytotoxicity are seed germination and plant growth bioassays [38]. We discovered that seed germination (%) following treatment with the degradation products was significantly higher than with the control dye (see Figure 6). Furthermore, when compared to the control MG, the degradation products were shown to be nontoxic in *B. cereus*, *S. aureus*, *M. luteus*, *Streptococcus sp.*, and *E. coli.* Finally, the cytotoxicity assay verified that the decolorized MG dye degradation products exhibited a decreased toxic effect. 

## 4. Material and Methods

### 4.1. Chemicals, Dye, and Microorganisms

All chemicals used were of analytical grade and were purchased from Oxoid. MG dye was obtained from Hi Media, Mumbai, India. All pathogenic strains of *S. aureus* ATCC 29737, *E. coli* ATCC 8379, *C. albicans* ATCC 60193, and *A. niger* ATCC 16404 were collected from Microbial Resource Center at the Faculty of Agriculture, Ain Shams University, Cairo, Egypt. Frying oil waste was collected from kitchen wastes. It has the following composition (9-Octadecenoic acid (Z)-, methyl ester; 9-Octadecenoic acid (Z)-, methyl ester; 11-Octadecenoic acid, methyl ester; 6-Octadecenoic acid, methyl ester (Z); 11-Octadecenoic acid, methyl ester; 6-Octadecenoic acid, methyl ester (Z); 6-Octadecenoic acid, methyl ester (Z)-; 11-Octadecenoic acid, methyl ester and Octadecenoic acid, methyl ester (Appendix A).

### 4.2. Isolation of Malachite Green-Biodegrading Streptomyces Isolates

Twenty soil samples were collected from planted soil at a depth of 15 cm near an industrial water effluent in Cairo, Egypt, packed into sterile plastic bags, transported to the lab at the Microbial Inoculant Center (MIC), Faculty of Agriculture, Ain Shams University, Cairo, Egypt, and stored at 4 °C for further investigation, as illustrated in Figure 14.

### 4.3. Culture Media

The indicator medium casein starch nitrate agar from Oxoid was supplemented with nalidixic acid (20 μg/mL) and nystatin (100 μg/mL) to inhibit bacteria and fungi, respectively, and 100 mg/L malachite green dye was used to isolate and maintain the Streptomyces isolates; in addition, it was used for optimizing the MG removal studies. Muller–Hinton agar medium was used for the detection of microbial toxicity with the following composition (g/L): glucose, 10.0; beef extract, 3.0; peptone, 5.0; agar, 20.0. The pH was adjusted to 7.0. All media were autoclaved at 121 °C for 20 min at 1 atm and stored at 4 °C for further investigation.

### 4.4. Standard Inoculum

Streptomyces isolates were grown before each experiment by inoculating 100 mL of casein starch nitrate broth with a loop of 72-day-old culture followed by incubation at 30 °C in a rotary shaker at 150 rpm. Bacterial cells in the late logarithmic growth phase were collected and centrifugated for 10 min at 8000 rpm and 4 °C. The supernatant was discarded, and the pelleted cells were collected and washed three times using 0.9% NaCl for further investigation.

### 4.5. Screening for Laccase Enzyme Production

Vogel’s mineral media (VMM) agar (pH 5.6) containing guaiacol (di-methoxy phenol) (Sigma) as substrate Vogel; Coll et al. [39,40] was prepared and poured into Petri dishes. Using a sterile cork-borer with a diameter of 6 mm, the plates were welled and filled with 0.1 mL of the actinobacterial cultures and then incubated at 30 °C for 120 h. The isolate with the largest orange or brownish zone was selected for further study.

### 4.6. Genomic Identification of the Selected Isolate Using 16S rRNA Gene Sequencing

The isolate’s genomic DNA was isolated and polymerase chain reaction (PCR) was utilized to amplify the 16S rRNA gene sequence using two universal primers (F1: 5′AGAGTTT (G/C) ATCCTGGCTCAG 3′ and R1 5′ ACGG (A/C) TACCTTGTTACGACTT 3′). The purification of the PCR product was completed using a QIAquick Gel Extraction Kit (Qiagen, Hilden, Germany). The genomic 16S rRNA gene sequencing of the purified PCR product was conducted by Macrogen Inc., South Korea. The sequence reads were trimmed and assembled using BioEdit version 7.0.4, and the virtually variable region of the 16S rRNA gene sequence was successfully aligned using the clusterW software version 4.5.1. The bootstrap values (%) were considered from 1000 resamplings. BLAST searches were completed using the NCBI server, according to the method described by Al-Dhabi and Esmail [41]. Using the ten 16S rRNA sequences from the NCBI gene bank that were most related to *S. exfoliatus*, a phylogenetic tree was generated using the neighbor-joining cladogram algorithms in the MEGA 11 program [42,43].

### 4.7. Growth Kinetics of S. exfoliatus

During the logarithmic phase, the specific growth rate (μG), doubling time (td), and multiplication rate (MR) were calculated using the following equations (Maier and Pepper, 2015): Specific growth rate (μG) (h − 1) = (ln X − ln X0)/(t − t0) where X is the amount of growth expressed by cell dry weight (g) after t time (t) and X0 is the amount of growth expressed by cell dry weight (g) at the start time (t0). Doubling time (td) = Ln (2)/μG. Multiplication rate (MR) = 1/td. 

### 4.8. Statistical Screening of Physical and Nutritional Factors Affecting Dye Removal by A. flavus NRRL3357 Using Plackett–Burman Design (PBD) 

Plackett–Burman design (“Design Expert® 12” Stat-Ease, Inc.; Minneapolis, MN, USA) was used to investigate the synergistic effects of nutritional and physical factors in a set of 18 trial runs measuring the decolorization percentage (%). Two carbon sources (mannitol and food oil waste), two nitrogen sources (ammonium nitrate and ammonium chloride), magnesium sulfate, and different dye concentrations were selected for the screening process. The studied physical factors were pH, agitation speed, incubation temperature, inoculum size, and incubation period. All eleven aspects were prepared at two levels (high and low), as shown in (Table 2). Color reduction% refers to the procedure for removing color (%R), and was calculated according to Casieri et al. [44] as follows:% R = 100 (A_0_ − A_t_)/A_0_
where A_0_ is the absorbance level at the initial dye concentration and A_t_ is the absorbance level after decolorization at time t.

All experimental trials were carried out in triplicate. Fisher’s test was used to calculate the statistical analysis of variance (ANOVA) to determine the effect of all tested parameters on the decolorization process. A Student’s *t*-test was used to determine the significance of each variable with 95% confidence levels by measuring the F-value, P-value, main effect, standard deviation, and coefficient of determination (R^2^). 

The Plackett–Burman design was based on the first-order model that was determined by Equation (1):Y = B_0_ + ƩB_i_x_i_(1)
where Y is the predicted response (decolorization percentage), B_0_ is model intercept, and Bi represents variable estimates.

The effect of each variable was determined by Equation (2):E(X_i_) = 2 (ƩM_i+_ − ƩM_i−_)/N(2)
where E(X_i_) is the tested variable effect; M_i+_ and M_i−_ represent the decolorization percentage for each trial where the variable (X_i_) was studied at high and low concentrations, respectively; and N is the number of trials.

The standard error (SE) was determined as the square root of the response variable, and the significance level (*p*-value) of each concentration effect was determined using Student’s t-test, as shown in Equation (3) below: t(X_i_) = E(X_i_)/SE(3)
where E(X_i_) is the variable (X_i_) effect.

### 4.9. Optimization of the Most Efficient Factors for Dye Decolorization Using Central Composite Design (CCD)

According to the results obtained using PBD, five factors were selected as the main significant factors (food oil waste, magnesium sulfate, dye concentration, pH, and inoculum size) in 32 trials to process parameter probabilities. The experiment was designed, data were analyzed, and a quadratic model was built using the “Design Expert” software (Version 12, Stat-Ease Inc.; Minneapolis, MN, USA).

Fisher’s test and statistical analysis of variance (ANOVA) were used to determine the effect of the physical and nutritional factors on the decolorizing percentage (%). A Student’s t-test was used to calculate the significance of the factors with 95% confidence levels. The model involved calculating each *F*-value, *p*-value, standard deviation, and coefficient of determination (R^2^). Decolorization percentage (%) was used as the dependent variable or response after completing the experiments (Y). Multiple regression analysis was used to fit a second-order polynomial (Equation (4)) to the results as follows:Y = β0 + β1A + β2B + β3C + β4D + β5E + β11A2 + β22B2 + β33C2 + β44AD2 + β55E2 + β12AB + β13AC + β14AD + β15AE + β23BC + β24BD + β25BE + β34CD + β35CE + β45DE(4)
where Y, is the predicted response of the decolorizing percentage (%); β0 is the intercept; β1, β2, β3, β4, and β5 are linear coefficients; β11, β22, β33, β44 and β55 are squared coefficients; and β12, β13, β23, β14, β15, β23, β24, β25, β34, β35, and β45 are interaction coefficients. Contour plots (3D) and response surface curves were employed to show the interaction between the significant variables in the decolorization process.

### 4.10. Phytotoxicity Toxicity of Malachite Green Dye before and after Biodegradtaion

#### 4.10.1. Broccoli (*Brassica oleracea* var. *italic*) and Lettuce (*Lactuca sativa* var. *capitata*) Seeds

To determine the phytotoxicity of the dye, the method described by Asses et al. (2018) [45] was applied using broccoli (*Brassica oleracea* var. *italic*) and lettuce (*Lactuca sativa* var. *capitata*) seeds, as they are common in agriculture. Ten seeds of each plant were placed in a Petri dish and watered daily with 3 mL of MG dye solution (100 mg/L) and biodegraded MG dye individually; distilled water was used as a control treatment. All plant seeds were incubated at ambient temperature for seven days, and the experiments were carried out in triplicate. To determine the phytotoxicity, germination percentage was calculated after seven days of germination as follows: Dye phytotoxicity = (Total number of germinated seeds/Total number of sowed seeds) × 100.

#### 4.10.2. Vicia Fabae Beans

Vicia fabae was grown in plastic pots (15 cm in diameter) filled with a mixture of sand and clay soil at a ratio of 1:1 (*v*/*v*); each pot was sown with a singular seed. All plants were watered daily and fertilized once a week with 0.1% NPK (20:20:20). The plants were kept under normal greenhouse conditions for the MG treatments. The treatments were divided into three groups according to irrigation. The first group (control) was irrigated using distilled water daily up to six weeks, the second group was irrigated with MG before treatment daily for six weeks, and the third group was irrigated with biodegraded MG daily for up to six weeks. The plant growth parameters, including shoot length, root length, and number of leaves, were recorded. All trials were carried out in triplicate, and the average mean of all measurements was calculated.

### 4.11. Microbial Toxicity

The well diffusion method was used to test MG toxicity in some microbial strains, as described by the CLSI [46]. Briefly, Muller–Hinton agar was prepared and poured into Petri dishes. The Petri dishes were inoculated with 0.1 mL of each strain culture containing 106 CFU/mL (*S. aureus ATCC* 29737, *E. coli* ATCC 8379, *C. albicans* ATCC 60193, and *A. niger* ATCC 16404) and welled using a sterilized cork-borer with a diameter of 7 mm; each well was filled with the degradation products of MG. Incubation was carried out at 37 °C for 5 d and the inhibition zones were recorded and expressed as zone diameters in cm.

### 4.12. Cytotoxicity

Human skin fibroblast (HSF) cells were obtained from Nawah Scientific Inc. (Mokatam, Cairo, Egypt). The cells were maintained in DMEM media supplemented with 100 mg/mL of streptomycin, 100 units/mL of penicillin, and 10% heat-inactivated fetal bovine serum in a humidified 5% (*v*/*v*) CO_2_ atmosphere at 37 °C. Cell viability was assessed using an SRB assay [47]. Aliquots of 100 μL of the cell suspensions (5 × 10^3^ cells) were placed in 96-well plates and incubated in complete media for 24 h. The cells were treated with another 100 µL aliquot of media containing drugs at various concentrations. After 72 h of drug exposure, the cells were fixed by replacing the media with 150 μL of 10% TCA followed by incubation at 4 °C for 1 h. The TCA solution was removed, and the cells were washed 5 times with distilled water. Aliquots of 70 μL SRB solution (0.4% *w*/*v*) were added, and the plates were incubated in a dark place at room temperature for 10 min. The plates were washed 3 times with 1% acetic acid and allowed to air-dry overnight. Then, 150 μL of TRIS (10 mM) was added to dissolve the protein-bound SRB stain; the absorbance was measured at 540 nm using a BMG LABTECH®-FLUOstar Omega microplate reader (Ortenberg, Germany). Cell viability was represented as the percentage of control cell viability, which was set as 100%.

### 4.13. Identification of the Biodegradation Products of Malachite Green Dye

The metabolites and intermediates produced by *S. exfoliatus* during MG dye degradation were investigated using two chromatographic methods.

#### 4.13.1. High-Performance Liquid Chromatography (HPLC)

The degraded MG products present in the cell-free supernatant were extracted using centrifugation at 8000 rpm for 20 min. The extract was dried over anhydrous Na_2_SO_4_ using a rotary evaporator, then dissolved in methanol [48]. HPLC was performed using an Agilent 1100, which is composed of two LC-pumps, a UV/Vis detector, and a 265 nm C18 column (125 mm × 4.60 mm, 5 µm particle size). Chromatograms were obtained and analyzed using the Agilent ChemStation. The operation conditions of the mobile phase were a gradient of methanol and water with 40% methanol in water for two minutes with flow rate of 0.5 mL/min, followed by a 10 min linear gradient to 95% methanol; this gradient was held for an additional 5 min with the same flow rate at a temperature of 35 °C [18].

#### 4.13.2. Gas Chromatography-Mass Spectrometry (GC-MS)

Before the GC-MS assay, the degraded MG products present in the cell-free supernatant were extracted using centrifugation at 8000 rpm for 20 min. The extract was dried over anhydrous Na2SO4 using a rotary evaporator, then dissolved in methanol [48]. The extract was further analyzed using GC–MS to investigate the intermediate products formed during the biodegradation of MG by *S. exfoliatus,* adopting the method of Mamoun et al. [49]. The chemical composition of the samples was determined using a TRACE GC-TSQ mass spectrometer (Thermo Scientific, Austin, TX, USA) with a TG–5MS direct capillary column (30 m × 0.25 mm × 0.25 µm film thickness). The column oven temperature was initially held at 50 °C and then increased by 5 °C/min to 250 °C, where it was held for 2 min. Then, the temperature was increased to the final temperature of 300 °C at 30 °C/min, before holding for 2 min. The injector and MS transfer line temperatures were kept at 270 and 260 °C, respectively. Helium was used as a carrier gas at a constant flow rate of 1 mL/min. The solvent delay was 4 min, and 1 µL of diluted sample was injected automatically using an Autosampler AS1300 coupled with GC in the split mode. EI mass spectra were collected at an ionization voltage of 70 eV over an *m*/*z* range of 50–650 in full scan mode. The ion source temperature was set at 200 °C. The components were identified by the comparison of their mass spectra with those from the WILEY 09 and NIST 14 mass spectral databases.

## 5. Conclusions

The actinobacterium *S. exfoliatus* biodegraded MG dye using a low-cost medium with the following optimal conditions: mannitol, 5.00 g/L; frying oil waste, 5.00 mL/L; magnesium sulfate, 0.25 g/L, ammonium nitrate, 1.00 g/L; ammonium chloride, 1.00 g/L; MG dye, 0.05 g/L; pH, 7; inoculum size, 10%; and incubation at 30 °C for 120 h. The metabolites of the biodegraded dye were found to be nontoxic to the studied broccoli seeds, lettuce seeds, faba bean plants, bacterial pathogen strains, and human cell lines. Moreover, the phytotoxicity test confirmed the absence of toxicity after the degradation process, indicating the efficiency of *S. exfoliatus* in minimizing this hazardous dye’s risk by lowering the toxicity level. These findings suggest that *S. exfoliatus* has the potential for the mycoremediation of dyes in grey water.

## Figures and Tables

**Figure 1 molecules-27-06456-f001:**
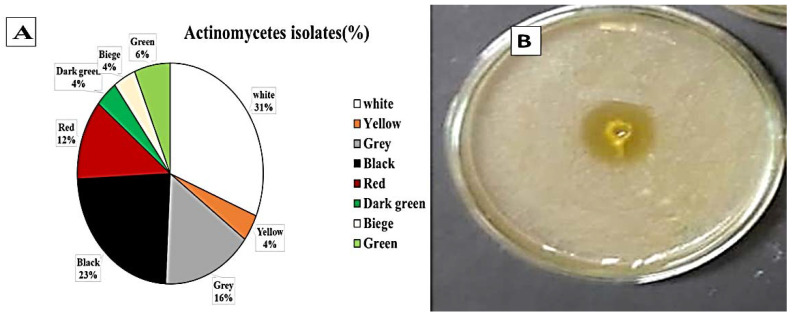
Morphological characteristics of laccase producing actinomycetes in soil samples. (**A**) colony color of Actinomycetes isolated from soil near industrial water effluent, Cairo, Egypt. (**B**) the most efficient producing laccase enzymes *Streptomyces* st 45 isolates on VMM agar plates incubated at 30 °C for 5–7 days indicating orange zone around well filled with 0.1 mL of actinomycetal culture.

**Figure 2 molecules-27-06456-f002:**
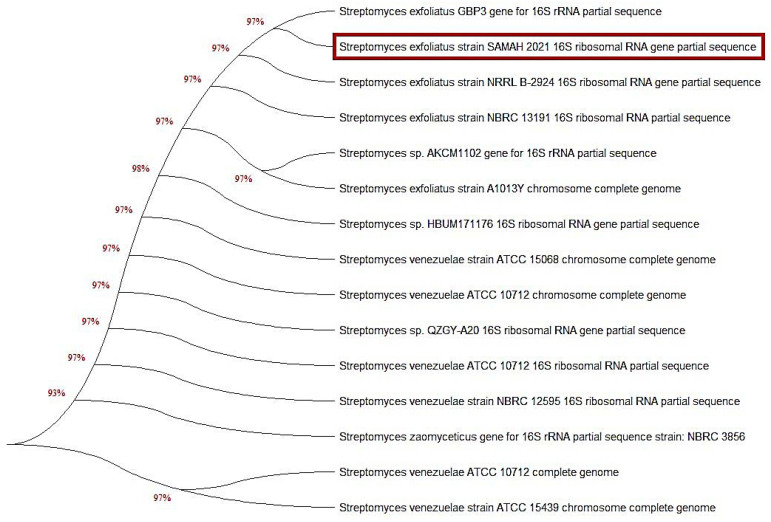
A phylogenetic neighbor-joining tree based on the 16S rRNA gene sequence of the most potent dyes-degrading strain *S. exfoliates* (marked in orange color) was assembled. The DNA sequence in the current study was deposited on the GenBank under accession number; OL720220 (*S. exfoliatus*). Utilizing MEGA X program, phylogenetic analysis was performed using the neighbor-joining algorithm.

**Figure 3 molecules-27-06456-f003:**
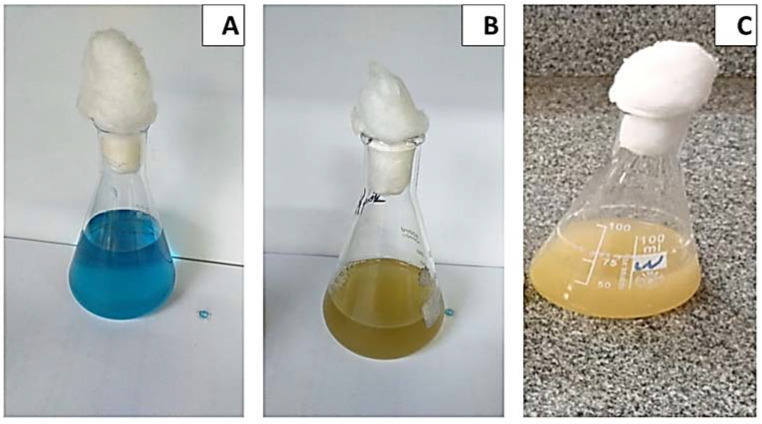
Effect of incubation time on growth kinetics and MG decolorization using *S. exfoliatus* incubated at 30 °C for 144 h. (**A**) MG color at zero time of incubation 30 °C for 144 h. (**B**) MG color at 72 h of incubation 30 °C for 144 h. (**C**) MG color at 144h of incubation at 30 °C for 144 h.

**Figure 4 molecules-27-06456-f004:**
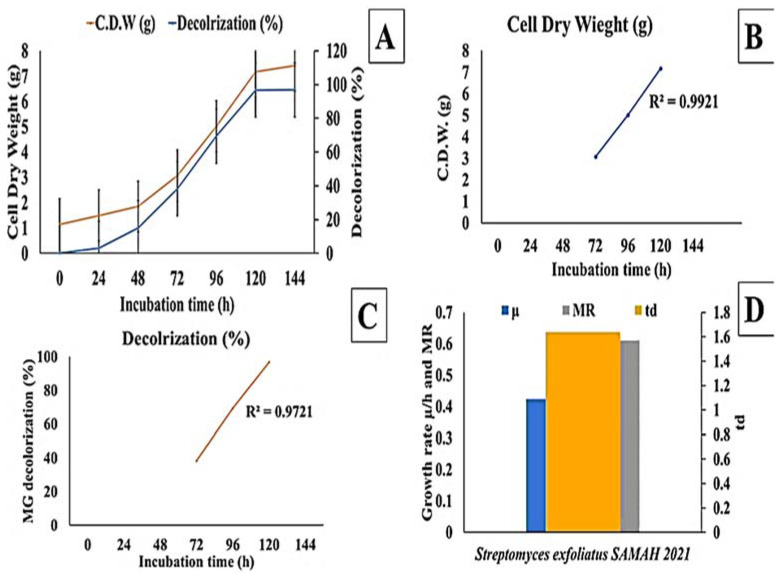
Effect of incubation time on MG degradation using *S. exfoliatus* incubated at 30 °C for 144 h. (**A**) decolorization (%) and dry cell weight (g) against time. (**B**) Cell dry weight (g) during the exponential phase of *S. exfoliatus*. (**C**) decolorization (%) during exponential phase of *S. exfoliatus*. (**D**) specific growth rate (µ/h), multiplication rate (MR) and doubling time (td) during the exponential phase of *S. exfoliatus*.

**Figure 5 molecules-27-06456-f005:**
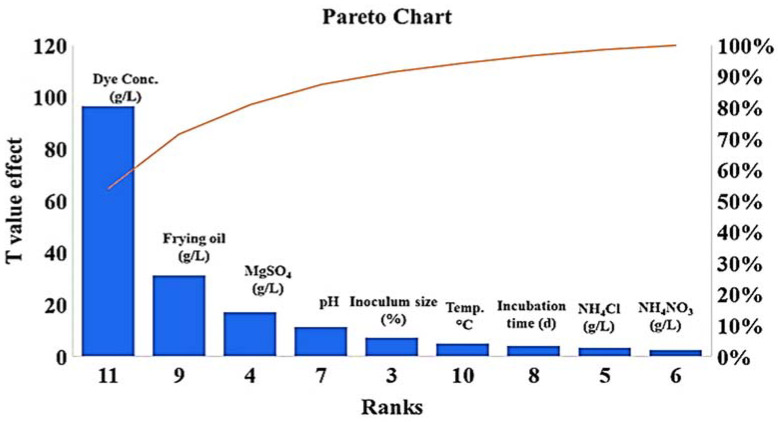
Main five significant factors affecting MG dye decolorization (frying oil waste, magnesium sulfate, dye concentrations, pH, and inoculum size) using *S. exfoliatus* strain.

**Figure 6 molecules-27-06456-f006:**
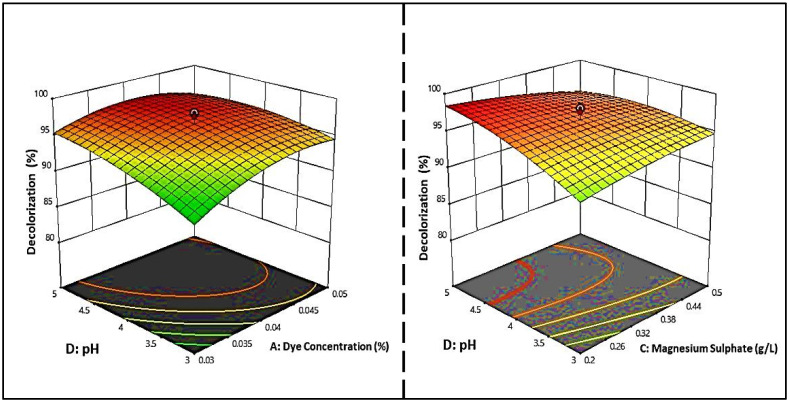
Three-dimensional response surface and two-dimensional contour plots illustrate the significant interaction of dye concentrations and pH, and MgSO_4_ on the decolorization of MG dye by *S. exfoliatus*.

**Figure 7 molecules-27-06456-f007:**
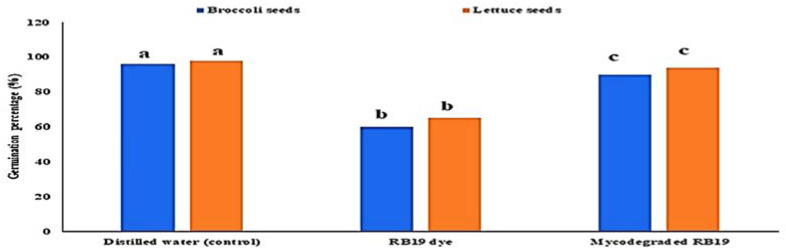
Effect of MG dye and its degraded metabolite products on the germination percentage (%) of broccoli (*Brassica oleracea var. italic*) and lettuce (*Lactuca sativa var. capitata*) seeds.

**Figure 8 molecules-27-06456-f008:**
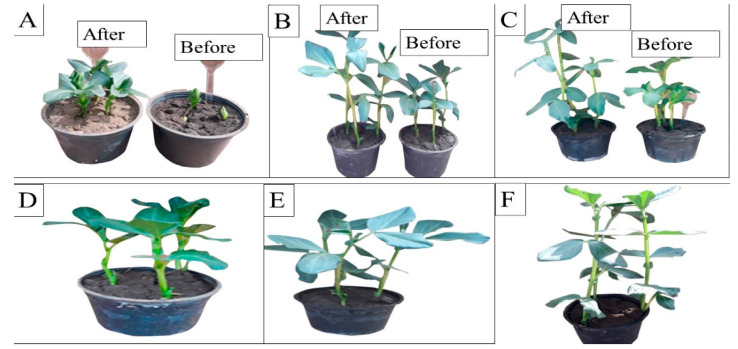
Phytotoxicity of MG and MG products on *Vicia faba* L. plants before and after biodegradation using *S. exfoliatus*. (**A**–**C**) irrigation with MG before and after MG degradation by *S. exfoliatus* during 6 weeks of planting, (**A**) after 4 weeks, (**B**) after 5 weeks, (**C**) after 6 weeks. (**D**–**E**) control treatment after 4 weeks, (**E**) control treatment after 5 weeks, (**F**) control treatment after 6 weeks.

**Figure 9 molecules-27-06456-f009:**
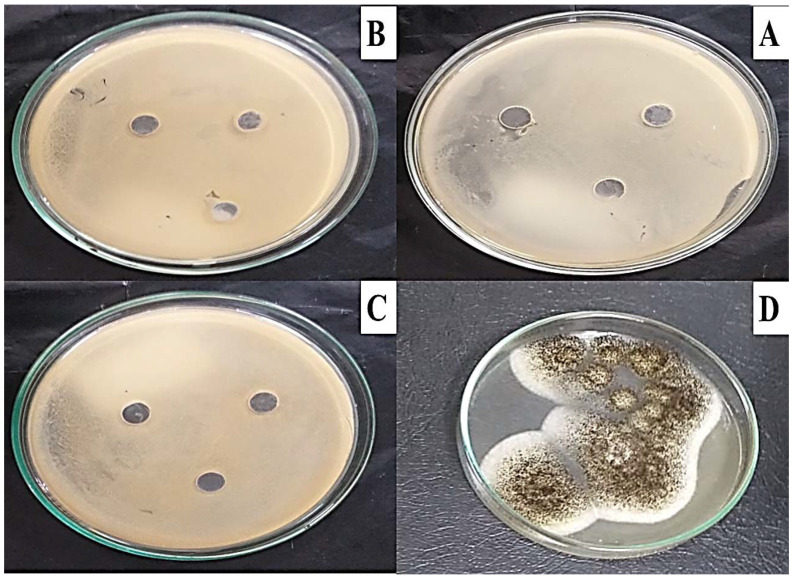
Microbial toxicity of MG biodegraded products against bacterial and fungal strains (**A**) *S. aureus* ATCC 29737, (**B**) *E. coli* ATCC 8379, (**C**) *C. albicans* ATCC 60193, and (**D**) *A. niger* ATCC 16,404 using the well diffusion method on Muller–Hinton agar indicating no inhibition growth for all strains.

**Figure 10 molecules-27-06456-f010:**
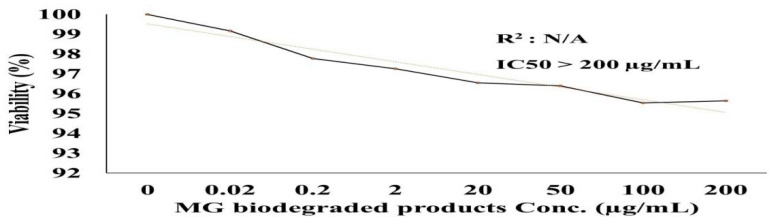
Cytotoxicity of MG and MG biodegraded products by *S. exfoliatus* strain on normal HSF cells maintained in DMEM media supplemented with 100 mg/mL of streptomycin, 100 units/mL of penicillin and 10% of heat-inactivated fetal bovine serum in humidified 5% (*v*/*v*) CO_2_ atmosphere incubated at 37 °C. (**A**) control treatment illustrates normal adherent cells. (**B**) cytotoxicity of MG by products show 96% viability at 200 µg/mL with few damaged cells illustrated by a reduction in cell adhesion.

**Figure 11 molecules-27-06456-f011:**
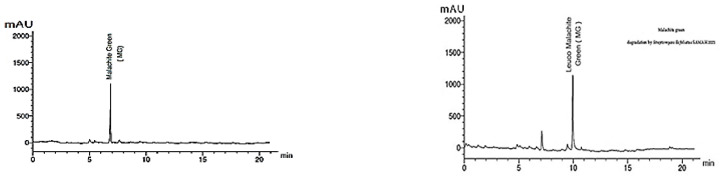
HPLC chromatogram of MG dye and its degraded metabolite products.

**Figure 12 molecules-27-06456-f012:**
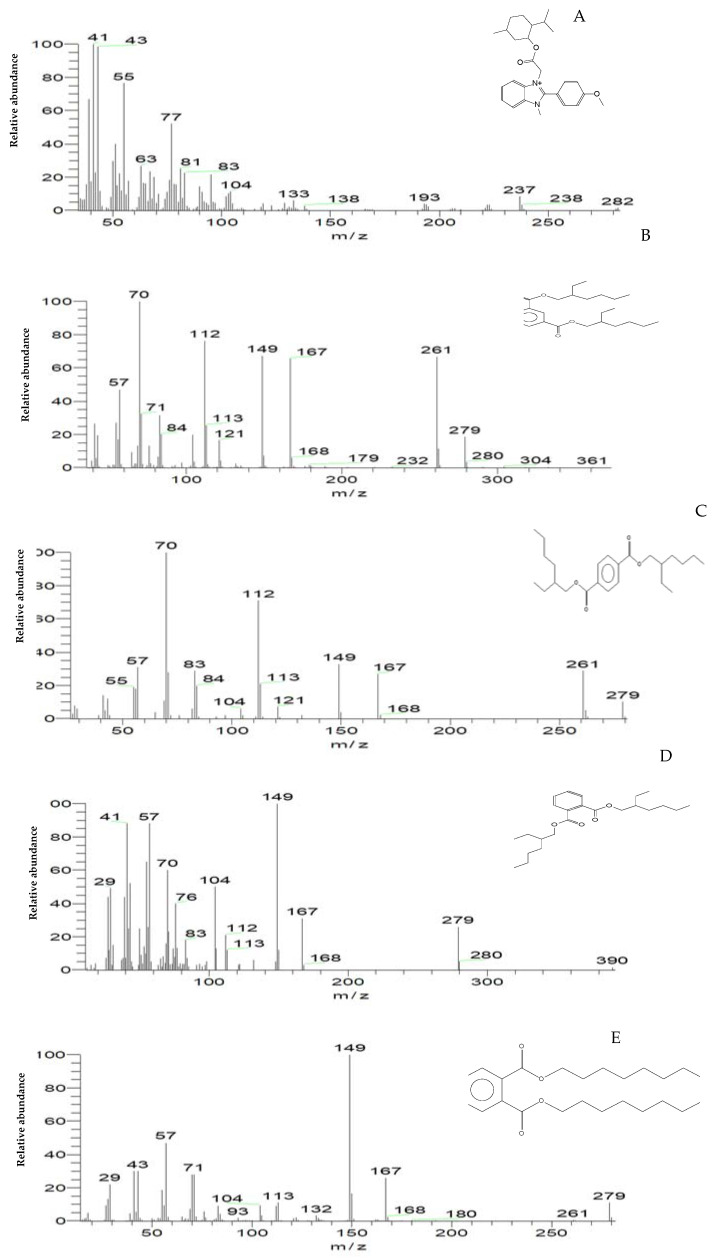
The mass spectrum of intermediates of the MG degradation identified by GC–MS analysis. (**A**) Di(tert-butyl)(2-phenylethoxy)silane (retention time 4.05); (**B**) 1,3-Benzenedicarboxylic acid, bis(2-ethylhexyl) ester (retention time 36.52); (**C**) 1,4-Benzenedicarboxylic acid, bis(2-ethylhexyl) ester (retention time 36.52); (**D**) 1,2-Benzenedicarboxylic acid (retention time 36.52); (**E**) Di-n-octyl phthalate (retention time 36); (**F**) 1,2-benzenedicarboxylic acid, dioctyl ester (retention time 36.52).

**Figure 13 molecules-27-06456-f013:**
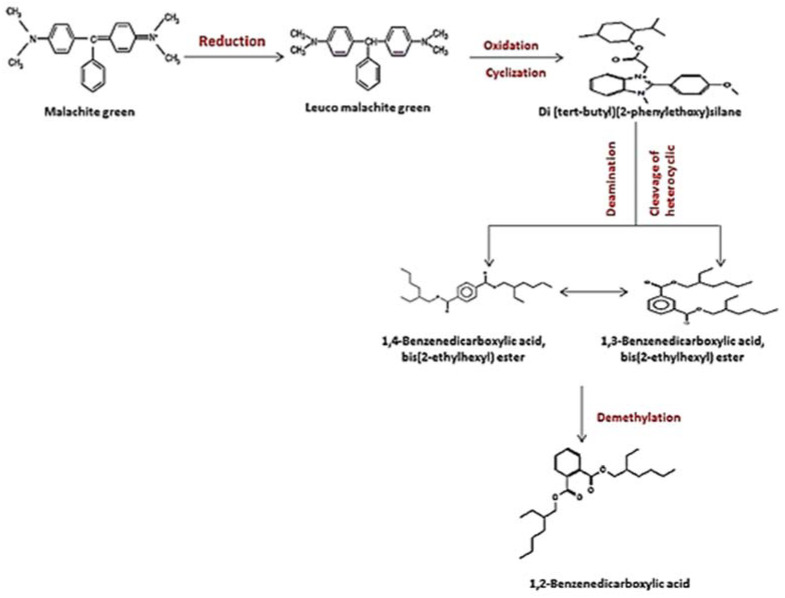
Suggested pathwasy for MG degradation by *S. exfoliatus* based on the results of HPLC and GC–MS analysis.

**Figure 14 molecules-27-06456-f014:**
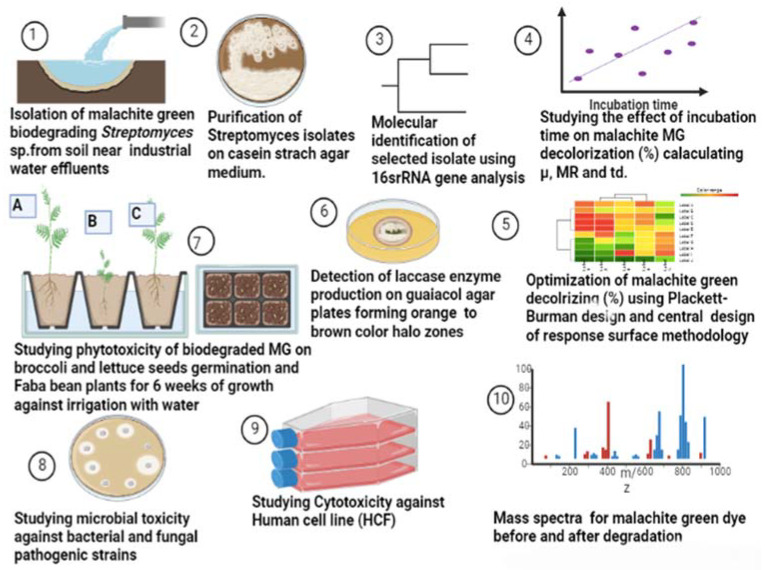
Flow diagram illustrating the work steps throughout this study.

**Table 1 molecules-27-06456-t001:** CCD matrix and ANOVA analysis against MG decolorization by *S. exfoliatus*.

**Runs**	**Levels**	**Variables**					**Decolorization** **%**
		**A**	**B**	**C**	**D**	**E**	
	Low	0.03	10.0	0.20	6.0	5.0	
	Mid	0.04	12.5	0.35	7.0	7.5	
	High	0.05	15.0	0.50	8.0	10.0	
1		0.04	12.5	0.35	7.0	7.5	97.56
2		0.03	10.0	0.20	6.0	10.0	82.73
3		0.04	12.5	0.35	8.0	7.5	98.10
4		0.04	12.5	0.35	5.0	7.5	91.26
5		0.05	10.0	0.20	6.0	5.0	93.29
6		0.04	12.5	0.65	7.0	7.5	93.76
7		0.03	15.0	0.20	6.0	5.0	90.71
8		0.05	15.0	0.50	8.0	10.0	94.96
9		0.04	12.5	0.35	7.0	7.5	95.97
10		0.04	12.5	0.35	7.0	7.5	95.97
11		0.05	15.0	0.20	6.0	10.0	91.54
12		0.03	10.0	0.50	8.0	10.0	93.99
13		0.04	17.5	0.35	7.0	7.5	94.37
14		0.05	10.0	0.50	6.0	10.0	93.61
15		0.04	7.5	0.35	7.0	7.5	95.06
16		0.04	12.5	0.35	7.0	7.5	97.72
17		0.04	12.5	0.35	7.0	2.5	95.97
18		0.04	12.5	0.35	7.0	7.5	95.44
19		0.02	12.5	0.35	7.0	7.5	86.06
20		0.03	15.0	0.50	6.0	10.0	91.55
21		0.05	15.0	0.50	6.0	5.0	93.29
22		0.03	15.0	0.50	6.0	5.0	94.18
23		0.05	15.0	0.20	8.0	5.0	97.16
24		0.05	10.0	0.50	8.0	5.0	92.58
25		0.03	10.0	0.50	6.0	5.0	89.86
26		0.03	15.0	0.20	8.0	10.0	96.43
27		0.04	12.5	0.35	9.0	7.5	93.16
28		0.04	12.5	0.05	7.0	7.5	95.21
29		0.06	12.5	0.35	7.0	7.5	92.28
30		0.03	10.0	0.20	8.0	5.0	96.62
31		0.04	12.5	0.35	7.0	12.5	93.31
32		0.05	10.0	0.20	8.0	10.0	96.83
Analysis of variance (ANOVA)							
	Model						
*F*-value	5.00	12.13	1.10	0.24	22.08	1.79	
*p*-value	0.004 *	0.005 *	0.316	0.632	0.001 *	0.208	
R^2^	0.9						
Std. Dev.	1.74						

A. Dye concentration(mg/L), B. frying oil waste (%), C. MgSO_4_ (g/L), D. pH level, and inoculum size (E) (mL/L). * Significant at 5% level (*p* < 0.05), *F* = corresponding level of significance, *p* = corresponding level of significance, Std. Dev. = standard deviation, and R^2^ = coefficient of determination.

**Table 2 molecules-27-06456-t002:** PBD design matrix and ANOVA analysis against MG decolorization by *S. exfoliatus*.

**Runs**	**Levels**	**Variables**											**Decolorization** **%**
		A	B	C	D	E	F	G	H	J	K	L	
	Low	0	0	0.1	1	1	6	0	25	5	96	0.05	
	High	5	5	0.2	5	5	8	150	35	10	120	0.50	
1		5.0	5.0	0.10	5	5	8	0	25	5.0	120	0.05	25.38
2		5.0	0.0	0.10	1	5	6	150	35	5.0	120	0.50	12.89
3		2.5	2.5	0.15	3	3	7	75	30	7.5	72	0.27	24.75
4		2.5	2.5	0.15	3	3	7	75	30	7.5	72	0.27	24.89
5		0.0	0.0	0.10	1	1	6	0	25	5.0	96	0.05	26.34
6		0.0	0.0	0.10	5	1	8	0	35	10.0	120	0.50	7.77
7		2.5	2.5	0.15	3	3	7	75	30	7.5	72	0.27	21.94
8		0.0	5.0	0.10	5	5	6	150	35	10.0	96	0.05	51.77
9		2.5	2.5	0.15	3	3	7	75	30	7.5	72	0.27	23.88
10		5.0	0.0	0.20	5	1	8	150	35	5.0	96	0.05	38.10
11		0.0	5.0	0.20	1	5	8	0	35	5.0	96	0.50	5.34
12		2.5	2.5	0.15	3	3	7	75	30	7.5	72	0.27	26.83
13		0.0	0.0	0.20	1	5	8	150	25	10.0	120	0.05	54.49
14		5.0	5.0	0.10	1	1	8	150	25	10.0	96	0.50	22.32
15		0.0	5.0	0.20	5	1	6	150	25	5.0	120	0.50	24.78
16		2.5	2.5	0.15	3	3	7	75	30	7.5	72	0.27	29.86
17		5.0	0.0	0.20	5	5	6	0	25	10.0	96	0.50	20.12
18		5.0	5.0	0.20	1	1	6	0	35	10.0	120	0.05	81.11
Analysis of variance (ANOVA)													
	Model												
*F*-value	18.17	2.47	7.42	17.12	3.41	2.64	11.54	4.18	1.75	31.3	5.14	96.54	
*p*-value	0.0004	0.160	0.030 *	0.004 *	0.107	0.148	0.012 *	0.080	0.234	0.001 *	0.058	<0.0001 *	
Main effect		4.90	8.50	12.91	−5.76	−5.07	−10.6	3.93	6.38	17.46	7.07	−30.66	
R^2^	0.96												
Std. Dev.	5.41												

A—mannitol (g/L); B—waste frying oil (ml/L); C—magnesium sulphate (g/L); D—ammonium nitrate (g/L); E—ammonium chloride (g/L); F—pH level; G—agitation speed (rpm); H—temperature (°C); J—inoculum size (%); K—incubation period (h); L—dye concentration (mg/L). * Significant at 5% (*p* < 0.05). *F*—corresponding level of significance; *p*—corresponding level of significance; Std. Dev.—standard deviation; R^2^—coefficient of determination.

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
