# Peer review of "Microbial Degradation, Spectral analysis and Toxicological Assessment of Malachite Green Dye by *Streptomyces exfoliatus"

_molecules, 2022, doi:10.3390/molecules27196456_

Round 1

Reviewer 1 Report

The manuscript describes aspects of microbial degradation of malachite green dye by Streptomyces exfoliatus SAMAH 2021 strain, a characterization of products of degradation as well as toxicity and cytotoxicity tests.

Unfortunately, I found some criticisms that preclude the acceptance of the manuscript in its present form.

Below are just a few things the Authors should keep in mind while revising their manuscript:

1) Introduction part lacks reference to the current state of knowledge of MG degradation.

2) Lines 75-77 - methods to decolorize dyes can be either physical, chemical, or biological. 

3) Lines 90-98 - the sentence is very long and therefore hard to follow; I recommend rephrasing.

4) Figures numbering starts with no. 2, not 1?

5) Fig. 2 - the caption of panel A is misleading; I recommend rephrasing. As for panel B - I recommend adding information (can be as a supplementary file) on measured diameters of the orange zone around isolates growth. Also missing is information on the controls used in this test.

6) Also, the topic of laccase has been treated very vaguely, and if the authors believe that these enzymes are essential in the degradation of MG, then at least for the selected strain, additional analyses should be performed, such as the determination of specific activity, or analysis of the protein/protein complex itself

7) Phylogenetic analysis was not done correctly. The Authors obtained results only by searches in the GenBank (NCBI) database for similar sequences. However, the database is full of sequences that are obtained from strains whose exact taxonomic affiliation was not determined because most depositors are not taxonomists. Thus, the search can easily be misleading. Authors should continue aligning their sequence individually with the sequences of the type strains of the species you found most similar in the GenBank search. It is not the case in the presented paper. Here you can find seven times the record for the same strain, which should not have happened! 

8) Fig. 4A – the caption should contain information on what is presented on panels A, B, and C.

9) Fig. 4B - – the caption should contain information on what is presented on panels A, B, C, and D.

It is unclear why the figures are 4A and 4B with panels A-D and not 4 and 5.

10) For PBD design - there is no information on the cut-off for high and low significance adopted. On what basis?

11) CCD design – There is no such parameter as standard division. It should be changed to “standard deviation.”

12) Lines 209-210 – in the formulation of %decolorization, the Authors could stick to the notation convention with the letters denoting the parameter in question

13) Line 269: “1 µg/mL, HSF cells had 99% of cell viability..” – this statement is inconsistent with the graph – please elucidate.

14) Line 272 - IC50 is the concentration of a drug or inhibitor needed to inhibit a biological process or response by 50%; therefore, it is unclear to me how was it calculated as 49.20, as for 200 ug/ml, estimated viability is at 96,5%? 

15) Fig.8 - the caption does not bring clear information on what is being presented. What cell line was used, what was the incubation time, and what is marked with red borders?

16) Lines 283-284 - broken thought in a sentence

17) Fig. 9 - the description of the vertical axis is missing

18) Lines 295-296 - broken thought in a sentence

19) Fig. 10 - the description of the vertical axis is unreadable

20) Materials and Methods, point 4.2 - it is not clear what ultimately constituted the sample, whether soil or water effluents (as presented in the figure)

21) Lines 495-497 – primers endings should be marked as 3’ or 5’

22) line 500 – “The sequence reads were trimmed and assembled using BioEdit version7.0.4, and then the virtually complete genomic sequence of the isolate was successfully 500 aligned using clusterW software version..” – it is not the complete seq. but a part of 16 rRNA seq.

23) line 504 The neighbor-joining cladogram – information on what was the reference parameters should be added

24) what do Authors mean by “Amount of growth”? Is it the conc. of dry biomass etc.?

25) Do the Authors know the composition of food oil waste, especially what other than carbon source elements it contains?

26) Page 19 Table 1 - what do the letters a, b, and c in the table description refer to?

27) Line 562 - what is the difference between F and P?

28) Line 663 – extraction is not equal to centrifugation; please correct the notation

29) HPLC analysis – please add temperature of the separation

Overall there are editing, punctuation, or stylistic errors in the text, I recommend their improvement, e.g. the notation of species names should be in italics, the notation of units should be standardized, and strain names should be unified through the text (once it is Samah 2021, in another place in the text SAMAH2021).

Author Response

Dear reviewer I

we would like to thank you so much for your time and effort. All your comments were very valuable for us to improve the overall quality of our manuscript. we have followed your suggestions and directions point by point. As well as, we have corrected the editing of English by MDPI editing service

This certificate will be sent as attached file to the editor

As for our responses to your comments, please find the attached file in this section

Regards

The authors

Reviewer 2 Report

The authors described the degradation of MG by actinomycetes strain. The study is very pertinent keeping in view the increasing pollutants in environment. The authors have used standard and advanced analytical techniques for analysis of MG degradation product.

Check the sequence of figures. Figure 2 is appearing first in the paper.

Figure 2 A- title is not self explanatory

Figure 4 A is further having 3 photos A, B and C but no details of the ABC is given in legends

Same is for figure 4 B. what are ABCD

Figure 5- units of T value effect ?

The manuscript need through editing by native English speaker. Many of the sentences in the MS hardly conveys the meaning.

Author Response

Dear reviewer II
We would like to thank you so much for your time and effort. All your comments were very valuable for us to improve the overall quality of our manuscript. we have followed your suggestions and directions point by point. As well as, we have corrected the editing of English by MDPI editing service
This certificate will be sent as attached file to the editor
As for our responses to your comments, please find the attached file in this section
Regards
The authors

Round 2

Reviewer 1 Report

Phylogenetic analysis is still done incorrectly - Authors should use not the sequences from GeneBank but align their sequence with the sequences of the type strains taken from known collections, such as NCYC, ATCC, DSM, of the species which they found most similar in the GenBank search. 

Food oli composistion should be added to the text. 

The other comments have been addressed.

Author Response

Dear reviewer I

We would like to thank you so much, Your comments help us to improve the overall quality of our manuscript. we appreciate your time and effort 

We have followed all your comments and directions, we hope the revised version of our manuscript  can meet now your high expectations 

As for our responses to your last comments, please find the attached file in this section

Regards

The authors

Reviewer 2 Report

The authors have incorporated all the suggestions and the MS is in acceptable condition

Author Response

Dear reviewer II

Dear reviewer1I,

We would like to thank you for your review and valuable comments of {Microbial Degradation, Spectral analysis and Toxicological Assessment of Malachite Green Dye by Streptomyces exfoliatus SAMAH 2021.} submitted to Molecules journal. After careful consideration of your comments and those of the other reviewers, this submission has been sent back for revision. I greatly appreciate the time and commitment that goes into each review.

Regards

The authors